# Microplastics Footprints in a High-Altitude Basin of the Tibetan Plateau, China

**Sansan Feng** [1,2], **Hongwei Lu** [2,*] **and Tianci Yao** [2]

1   School of Renewable Energy, North China Electric Power University, Beijing 102206, China; sansanf123@163.com
2   Key Laboratory of Water Cycle and Related Land Surface Process, Institute of Geographic Science and Natural Resources Research, Chinese Academy of Sciences, Beijing 100101, China; Tianciyao2018@163.com
*   Correspondence: luhw@igsnrr.ac.cn; Tel.: +86-10-64849311; Fax: +86-10-64851844

**Abstract:** Microplastics (MPs) are ubiquitous in the environment and have been drawing increasing attention; however, MPs' occurrence and behavior in remote areas are not well understood. In this study, we quantified and characterized MPs from surface waters and sediments in a remote area, namely the Tibetan Plateau, China. The samples were collected from the Lhasa River and the lower reaches of the Brahmaputra River to better understand MPs' sources to rivers of the Tibetan Plateau. MPs' concentrations in water and sediment were 735 items/m$^3$ and 51 items/kg, respectively, and the dominating MPs observed were fibers with size ranging from 100 to 500 μm. MP abundance increased nearly two-fold from upstream to downstream in the Brahmaputra River, associated with the inputs from downstream human activities and the inflows of tributaries (especially the Lhasa River). This study provides important bases for analyzing MPs migration processes in the plateau region.

**Keywords:** microplastic; water; sediment; Brahmaputra River; Tibetan Plateau





## 1. Introduction

Microplastics (MPs), the most frequently used definition of which is a type of plastic measuring less than 5 mm in size [1], have drawn increasing attention over the past few years due to their threats to the environment [2,3]. Evidence has shown that MPs could probably affect the health of organisms by multiple pathways, such as adherence, contact, ingestion, and through food chains [4,5]. MPs in the aquatic environment might be accidentally ingested by various organisms, including seabirds, fish, bivalves, and mammals [6,7]. Moreover, MPs are transferred through the food chain and ultimately enter the human body, resulting in a potential risk for human health [8,9]. In addition, MPs may act as a toxicological reservoir, adsorbing harmful substances, such as heavy metals and pharmaceuticals. MPs can consider a global environmental concern, which threatens the human health, given their pervasive and potential perniciousness [10,11]. Therefore, studying the MP source in the aquatic environment is important to prevent and minimize their environmental risk.

MP pollution was first considered a problem in the marine environment, and early studies on MPs mainly focused on the world's oceans [12,13]. Approximately $160 \times 10^6$ MP particles are being released into coastal water bodies per day through gray water [14,15]. The occurrence of MPs has been reported in the Atlantic, Pacific, and Indian Ocean [16–18], and a relatively higher MP abundance was also detected in the offshore environment and deep-sea sediments [19,20]. Subsequently, an increasing number of studies has reported the occurrence of MPs in freshwater environments worldwide, because rivers can act as important pathways for land-based MPs entering the ocean, and estuaries as transition zones between oceans and streams [21–24]. To date, MPs have been found in various aquatic environments on the terrestrial ecosystem, such as rivers and lakes [25–27]. The concentration of MPs in the Poyang Lake of China ranged from 5000 items/m$^3$ to 34,000 items/m$^3$ [28],

and Cahya et al. revealed high levels of MPs (i.e., 5850 item/m$^3$) in the Ciwalengke River, Indonesia [29]. In addition, the investigation of MPs has been expanded to remote areas [30–32]. Although MPs in the freshwater environment have been extensively studied, the research on MP distribution in remote areas is scarce, particularly in headwater regions.

Current studies on MPs in the freshwater environment mainly focus on areas at low altitude. However, few studies have mentioned the remote/high altitude regions [33–35]. The Tibetan Plateau is situated in Western China with a mean elevation over 4000 m; it is a remote environment that has received few systematic studies on MPs. Two previous studies have investigated MP composition and spatial distribution in the rivers of the Tibetan Plateau and proposed that domestic sewage might be a general source of MPs in the Tibetan Plateau [30,31]. It was found that the Brahmaputra River, as the longest plateau river, received substantial MP input from its tributaries in the region, particularly Lhasa River as the largest tributary. However, the two previous studies only focused on MP abundance of specific upstream and downstream sections of the Brahmaputra River independently, while a systematic comparison between upstream and downstream sections had not been performed. In addition, the effects of tributaries on MPs in the mainstream were not considered. Therefore, MP abundances and composition between the upper and lower reaches of the Brahmaputra River (before and after the Lhasa River) were studied further to better understand the occurrence and fate of MPs within the different river segments in a high-altitude basin of the Tibetan Plateau.

Here, water and sediment samples were acquired from the rivers, lakes, and agricultural channels in the lower course of the Brahmaputra River, the Lhasa River, and their vicinity to determine the characteristics of MPs and evaluate the impact of the counties' agglomerations on MP abundance in a high-altitude basin of the Tibetan Plateau, China. The occurrence of MPs in water and sediments at different sites was examined, as well as their composition, such as shape, size, color, and polymer type. We also assessed the change in MP abundance/composition during migration from upstream to downstream of the Brahmaputra River, and the multiple sources of MPs in the water were explored.

## 2. Materials and Methods

### 2.1. Sampling Sites

The study area (28°10′17″–31°38′28″ N, 90°0′46″–94°27′53″ E) is located at the central Tibetan Plateau (Figure 1). The Brahmaputra River, which is the longest plateau river in China, lies in the Tibet Autonomous Region (Figure S1). MP sources, such as domestic sewage, agriculture activities, and inflow of tributaries, in the upstream and the downstream of the Brahmaputra River were found along its 3848 m length. Lhasa is the capital of Tibet; it is also the most populous city in Tibet with a population of 860,000. Lhasa River, a major tributary in the middle reaches of the Brahmaputra River, passes through the city of Lhasa and then flows into the Brahmaputra River (Figure 1). Influenced by the Indian monsoon, the mean annual temperature in the study area is 5.3 °C, with an annual mean precipitation of 400–500 mm [36,37]. The study area is a typical farming and pastoral area, and agricultural facilities (mainly composed of plastic mulching and greenhouse) have been widely used in the Tibetan Plateau due to the cold conditions of the plateau. Plastic products were also widely used by residents on the Tibet Plateau, and a large number of tourists enter the plateau area. Hence, plastic waste has become an important source of MPs in the local natural environment.

A major objective of this study was to compare the MP abundances and compositions between the upper and lower reaches of the Brahmaputra River (before and after the Lhasa River). Hence, water samples were collected from the lower reaches of the Brahmaputra River (the data of upstream river were obtained from [31]), the Lhasa River, and surrounding water bodies. Obtaining sediments from some sampling sites was difficult due to the terrain conditions of the plateau; thus, only eight sediment samples were obtained in this study. The sampling sites include 14 sites, namely rivers ($n$ = 9), lakes ($n$ = 3), and agricultural canals ($n$ = 2), and the samples were collected in August 2020.

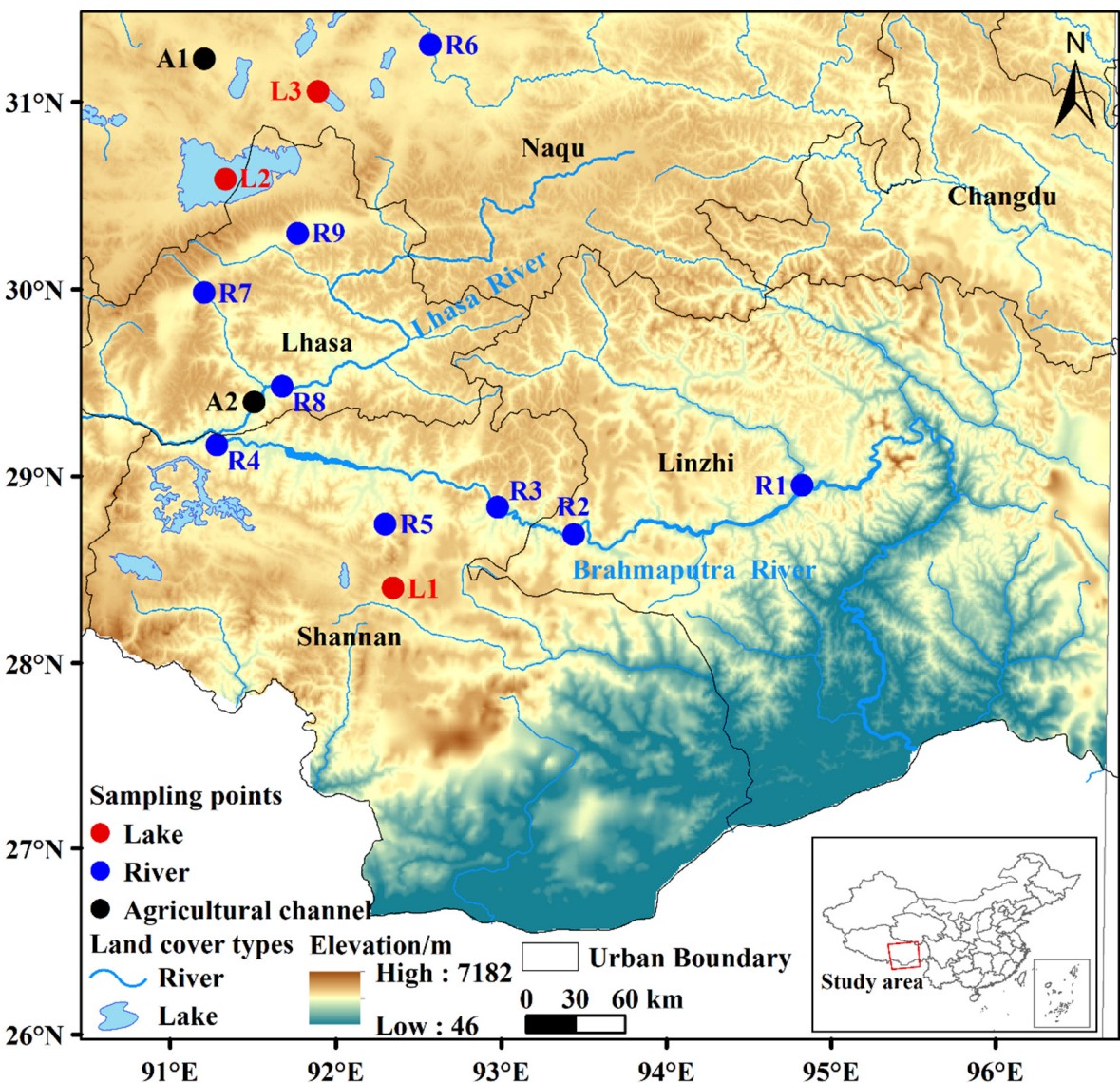

**Figure 1.** Locations of sampling sites in Tibetan Plateau.

*2.2. Sample Collection*

Surface water sampling was performed using a metal bottle (2 L) with a 5 m telescopic rod because no boat was available on the plateau, and only surface water was obtained in this study (top 10 cm). Many previous studies have used 20 L to obtain representative MP concentrations [38,39]. We collected 20 L of water as the sampling volume, considering the sampling workload and representativeness of samples. At each site, 20 L surface water (10 buckets) was sampled and then filtered through a metal screen (20 μm mesh size; 20 cm diameter) to obtain retained samples on the sieve; then, the samples were rinsed into a 250 mL glass bottle by mineral water (Table S1). We selected a longer distance to obtain triplicate samples due to the special natural conditions of the Tibetan Plateau, i.e., larger width of rivers and lakes. Thus, to obtain subsamples that are more representative when considering the spatial heterogeneity in water, three parallel samples, 20 m away from each other, were deployed and measured for quality control.

The sediment samples were obtained using a metal shovel from the shoreline of lakes and from the water line in rivers. Triplicate samples, 15 m away from each other within a 0.2 m × 0.2 m sampling area (top 5 cm), were collected at each site. The sediment was immediately transferred into an aluminum foil bag for storage. All samples were preserved in 4 °C within a portable cooler for further analysis.

### 2.3. Microplastic Extraction

Water samples were poured into a 500 mL glass beaker and treated with 30% $H_2O_2$ at 70 °C for 72 h to digest organic materials. At times, digestion time was adapted depending on the color (until they were clear in appearance). Then, the samples were filtered through glass GF/C filters (0.45 μm pore size; 47 mm diameter, Joan Lab, China) using a vacuum pump [40]. After filtration, the filter papers were placed in glass Petri dishes (52 mm diameter) prior to further analysis.

The sediment was placed on a glass pie tray and dried at 58 °C for 24 h. Then, the dried samples (150 g) were sieved through 5 and 2 mm meshes successively to remove larger debris (the plastic is singled out and counted) [41], and filtered samples were placed in a 1000 mL glass beaker. Density separation was used to extract MPs from sediments. A saturated $CaCl_2$ solution (900 mL, 1.5 g cm$^{-3}$) was added to each glass beaker and stirred for 0.5 h, before the mixed solution was left to settle for at least 24 h. Finally, the supernatant was transferred into a 250 mL conical flask. The process was replicated three times with the flotation solution changed to NaCl solution (1.2 g cm$^{-3}$) (See Supplementary Materials Text S1 for detailed reasons). After the degradation was completed, the solution in the beaker was treated similarly to the water samples.

### 2.4. Microplastic Identification

A stereomicroscope (Chongqing COIC Industrial Co., Ltd. UB100-CV320, Chongqing, China) was used to visually inspect the sample filters under the magnification of 40× and 100×. The images of all particles were obtained by a camera link to UopView software (UB100-CV320), and the longest length of particles was measured. MPs were classified according to the shape, size, color, and quantity. Furthermore, all suspected MPs on the filters were confirmed by micro-Raman spectroscopy (Gloucestershire, UK), and the spectra ranged between 3200 and 100 cm$^{-1}$ (incident laser: 535 nm; collection time: 0.5 s). All spectra were analyzed using a spectral library (Bruker), and particles with ≥80% similarity were regarded as MP polymers.

### 2.5. Quality Control

Several countermeasures were taken to prevent possible sample contamination with plastics. Sampling instruments (e.g., bottle, spatula, and metal screen) in the field were washed with mineral water prior to use, and the laboratory equipment was thoroughly cleaned with distilled water in the laboratory. Additional lab analysis of the mineral water used in the field showed no contamination with MP particles above 20 μm, i.e., the lower size limit of our investigation. During the extraction and analysis, the materials and instruments were covered with aluminum paper to avoid contaminations with air. In addition, cotton coats and gloves (nitrile) were worn to complete the experiment, and the entire process avoided contact with all plastic products. Blank samples that used deionized water were treated through the entire analysis procedure, and one item/filter paper, corresponding to 20 items/m$^3$, was observed on the blank filter.

### 2.6. Data Analysis

Statistical analyses were performed using SPSS (SPSS Inc., Chicago, IL, USA). All results for water and sediment samples are report as composite numbers of triplicates per location. To compare the significant differences among MP abundance in different segment areas, the analysis of variance (ANOVA) with *t*-test was analyzed and considered significant at *p*-value < 0.05. The figures were performed using Microsoft Excel 2017 (Microsoft Inc., Seattle, WA, USA), Origin 2016 (Origin Lab Inc., Northampton, MA, USA), and ArcGIS 10.0 (ESRI Inc., Redlands, CA, USA) software.

## 3. Results and Discussion

### 3.1. Abundance and Spatial Distribution of MPs

MPs were detected in all samples with abundance ranging from 188 items/m$^3$ to 1500 items/m$^3$ in water and 30 items/kg to 90 items/kg in sediment, respectively, showing a pervasiveness of MP pollutant in the basin. For the water samples, the mean MP abundance was $744 \pm 383$ items/m$^3$, which was comparable to that found in the Limpopo river (705 items/m$^3$) [42]. The highest MP concentration was found in the downstream section of the Brahmaputra River (R1; 1500 items/m$^3$); it was significantly higher than the mean MP abundance in the upstream section (338 items/m$^3$) [31]. This significant discrepancy between the two segments was mainly attributed to the different MP local sources. The concentration of MPs at L1 (188 items/m$^3$) was significantly lower than those of others located near an uninhabited grassland at the southern Tibetan Plateau without direct plastic sources. A comparison between this study and other studies that used a similar sampling approach was conducted to better understand the MP concentration in the basin. MP abundance in the water samples of the study area is considerably lower than many other regions as shown in Table 1. The mean MP abundance in the Pearl River, China (2274 items/m$^3$) was three times higher than that in our study [43], and the MP concentration in Lake Ontario, Canada was also approximately two to three orders of magnitude higher than our results [44]. The average MP abundance was $53 \pm 21$ items/kg in sediment samples; it was evidently lower than that detected in Pearl River, China (80–9897 items/kg) [43]. Nel et al. found a relatively low concentration of MPs ($160 \pm 139$ items/kg) in an urban river in South Africa (summer) [45], whereas MP concentration in the sediment of the Bizerte lagoon, Northern Tunisia ($7960 \pm 6840$ items/kg) [34] was considerably higher than in our study.

**Table 1.** Studies on MP abundance in different freshwater environments.

| Location | Size Range (μm) | Minimum Abundance (Items/ m$^3$) | Mean Abundance (Items/m$^3$) | Maximum Abundance (Items/ m$^3$) | Reference |
|---|---|---|---|---|---|
| Goulburn River, Australia | 20–5000 | - | 400 | - | [46] |
| Yangtze Estuary, China | 32–5000 | 160 | 4140 | 10,200 | [38] |
| Tibet Plateau, China | 20–5000 | 187.5 | 743.9 | 1500 | This study |
| Three Gorges Dam, China | 48–5000 | - | 4703 | - | [47] |
| Ox-Bow Lake, Nigeria | 20–5000 | 201 | - | 8369 | [48] |
| Pearl River, China | 20–5000 | 379 | 2274 | 7924 | [43] |
| Nakdong River, Korea | 20–5000 | 293 | - | 4760 | [49] |
| Lake Ontario, Canada | 10–5000 | 900 | 15,400 | - | [44] |

As shown in Figure 2, a remarkable difference in the average MP concentration between the river and lake water bodies was found in this study ($p < 0.05$). The MP abundance in rivers was significantly higher than that in the lakes. Previous studies have shown that domestic sewage discharge was a major input of MPs in the Tibetan Plateau [31]. Most river sampling sites were located in the lower reaches of urban areas and counties (e.g., R1, R2, R3, and R4). Thus, MP input from human activities near rivers is greater than near lakes. In addition, the MP abundance in the agricultural channels is higher than that in lakes, because the agricultural channels were close to the agroecosystems. High levels of MPs have been discovered in the farmland of the Tibetan Plateau; they were likely to flow in from the farmland into agricultural channels through runoff. The lakes are scenic spots and far from populated areas, which were different from the agricultural channels and some rivers. Hence, MP input into lakes from direct and diffuse sources is low compared to that in rivers and channels. Meanwhile, previous studies have demonstrated that the tourism industry might be a direct MP source in these lakes [31]. A similar distribution trend was also detected in sediments, implying similar sources of MPs in water and sediment. However, the MP concentration in the lake sediment was not significantly lower than those

in other types of sites ($p > 0.05$), because the lakes had an enclosed and static environment compared with others, thereby facilitating mostly gravitational settling of MPs.

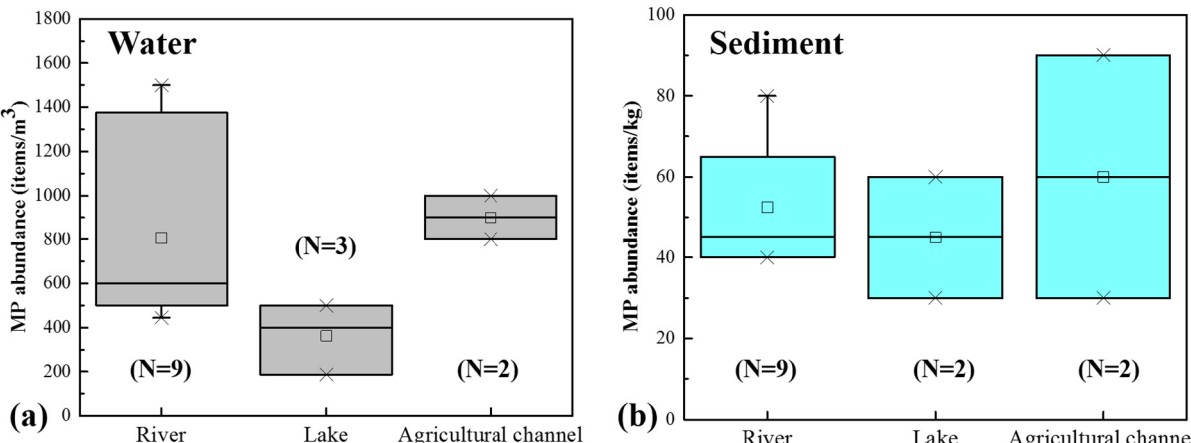

**Figure 2.** MP abundances within different environments. (**a**) MP abundances within different water bodies; (**b**) MP abundances within different sediments. (The little box represents the average value, and the horizontal line in the middle of the box represents the median).

### 3.2. Polymer Shape and Composition

Under the microscope, the typical microscopy images of MPs are as shown in Figure 3; they were classified into five morphology categories (film, fragment, fiber, foam, and granule). Among these shape types, fibers were dominant in all samples and accounted for 47% and 39% of the total MPs in water and sediment, respectively (Figure 4). Plastic fibers were followed by fragments (19%), films (14%), foams (13%), and granules (7%) in water and by fragments (30%), films (15%), foams (9%), and granules (7%) in sediment. The size of the MPs was classified into seven categories (1000–5000 μm, 750–1000 μm, 500–750 μm, 250–500 μm, 100–250 μm, 50–100 μm, and 20–50 μm). The MPs in the size category 100–500 μm were most abundant in the water and sediment samples, corresponding with previous studies with relatively dominant smaller MPs [50]. In surface water, the MPs < 500 μm accounted for 75%, and MPs > 1 mm only accounted for 9%. Similarly, MPs < 500 μm were abundant in the sediment and accounted for 86% of the total MPs. Significantly, the proportion of MPs < 100 μm was lower in water (34%) than in sediments (48%), probably because small MPs were propitious to float in water given their greater external surface area [51].

As shown in Figure 3, MPs were classified into seven colors, and the dominant color in the samples was transparent (water: 40%; sediment: 40%). In the water, the remaining colors were white (16%), gray (20%), green (9%), blue (7%), and yellow (2%). Meanwhile, the color distribution was considerably similar between the water and sediment (Figure 3c). The μ-spectroscopy analysis results showed 13 polymer types in the samples (Table S2). The most frequently identified polymer types were polyethylene (PE, 36%) in water but polypropylene (PP, 33%) in sediment. The remaining polymer types were PP (29%) > others (14%) > polystyrene (PS, 12%) > polyethylene terephthalate (PET, 9%) in water; the remaining MP polymer types in the sediment were PE (29%) > others (14%) > PS (13%) = PET (13%). The composition characteristics are important features, contributing to the identification of the potential sources of MPs. MP fibers were prevalent at most sampling sites, and laundry/domestic sewage may be a potential fiber source [51]. In addition, the breakdown of packaging materials, plastic containers, and cosmetics were the main source of fragments, and similar conclusions were reported in previous studies [52,53].

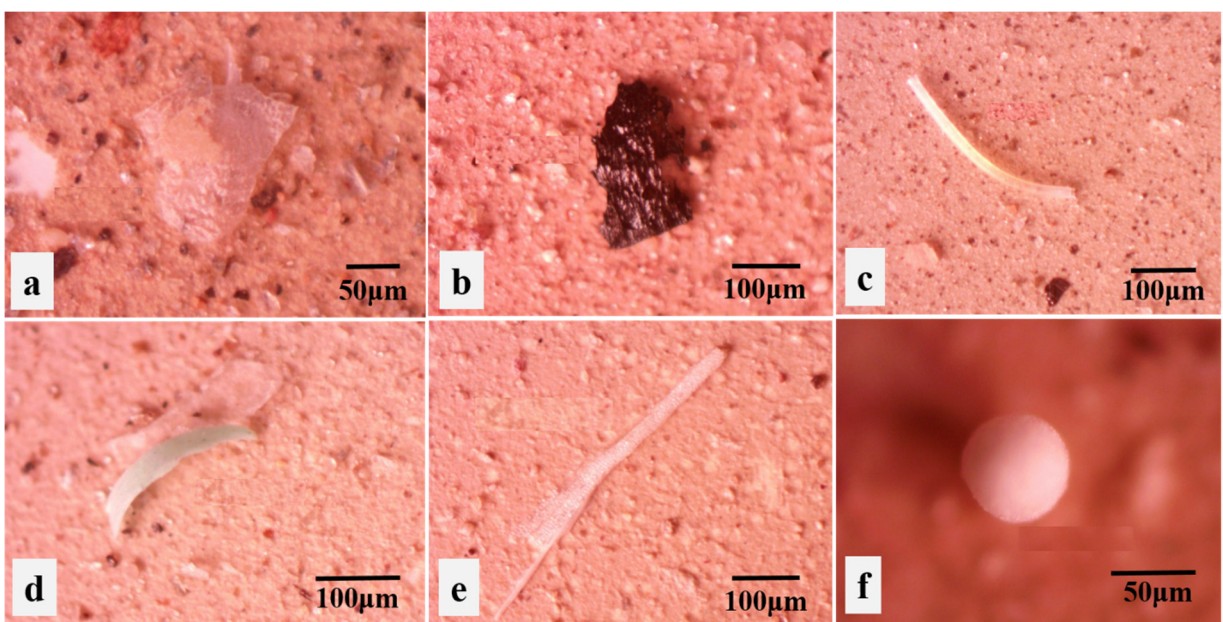

**Figure 3.** Typical microscopy images of MPs (films (**a**,**b**), fiber (**c**), fragment (**d**), foam (**e**), and granule (**f**)).

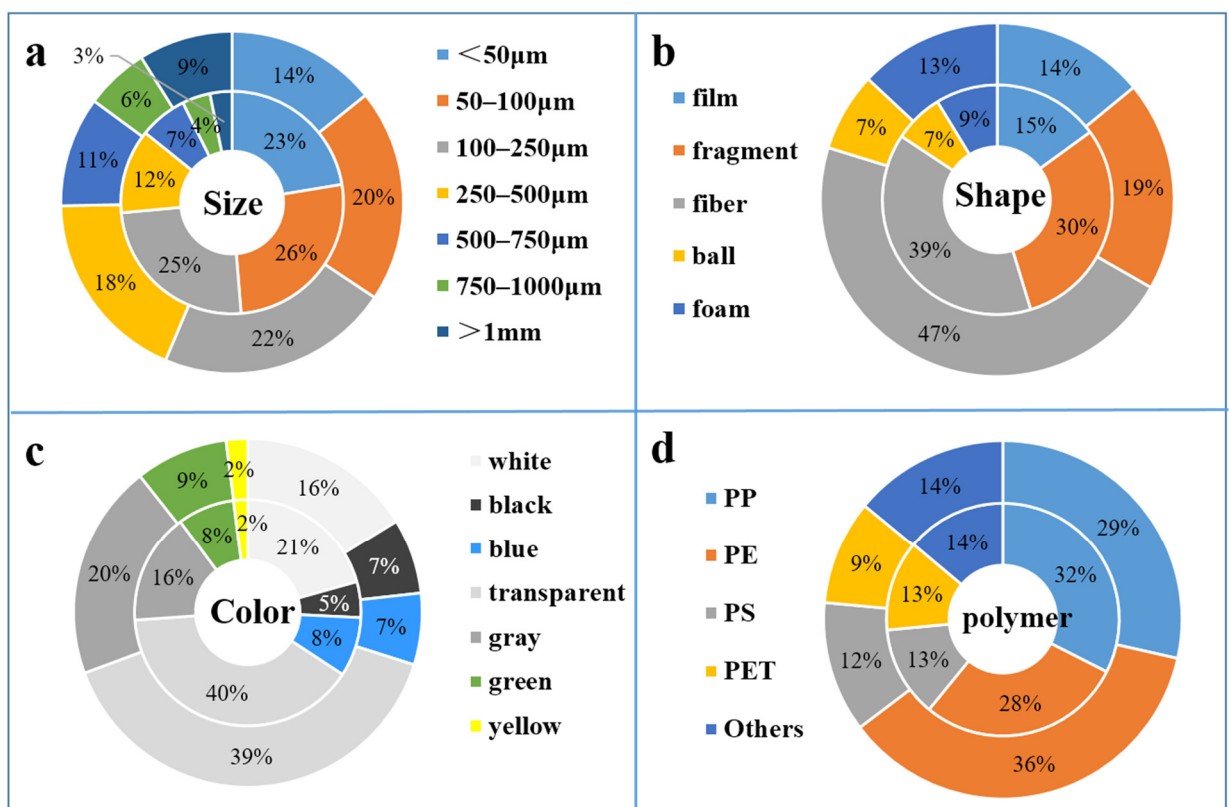

**Figure 4.** Size (**a**), shape (**b**), color (**c**) and polymer (**d**) distribution of MPs in samples (outer ring = water, inner ring = sediment).

Figure S2 shows that, for different water body types, fibers dominated in lakes, implying the shedding of fibers from clothing of visitors near the lakes. The most abundant color (other: 33%) of MPs in the agricultural channel(s) may be related to nearby agricultural activities. However, no significant difference in polymer and size composition was found between different water body types, and one reason may be the limited number of samples.

### 3.3. Comparison with Upstream Segments of the Brahmaputra River

The study area was located in the downstream sections of the Brahmaputra River, the longest plateau river in China, and MP abundances among different segments of the Brahmaputra River revealed diverse features that correspond to the respective sources. The MP concentrations between the studied upstream and downstream sections of the Brahmaputra River were compared, and a remarkable difference was observed among different segments, as shown in Figure 5 [31]. In general, lower MP abundances (100–533 item/m$^3$) were evident at rural sites of upstream reaches compared with the downstream sections near the counties (500–1500 item/m$^3$), and a similar trend was found in sediments based on limited samples. Feng et al. [31] previously showed significant correlation between MP concentrations and altitude, with higher altitudes corresponding to lower socioeconomic development and lower MP abundances and similar observations were made here. The altitude difference between the studied upstream and downstream section of the river was more than 1000 m, and upstream sampling sites were located in pastoral areas, whereas downstream sampling locations were located downstream of populated counties (e.g., Jiacha County and Lang County). Therefore, we hypothesize that the significantly higher MP levels in the downstream sections are mostly a result of the input of plastic waste from downstream populations.

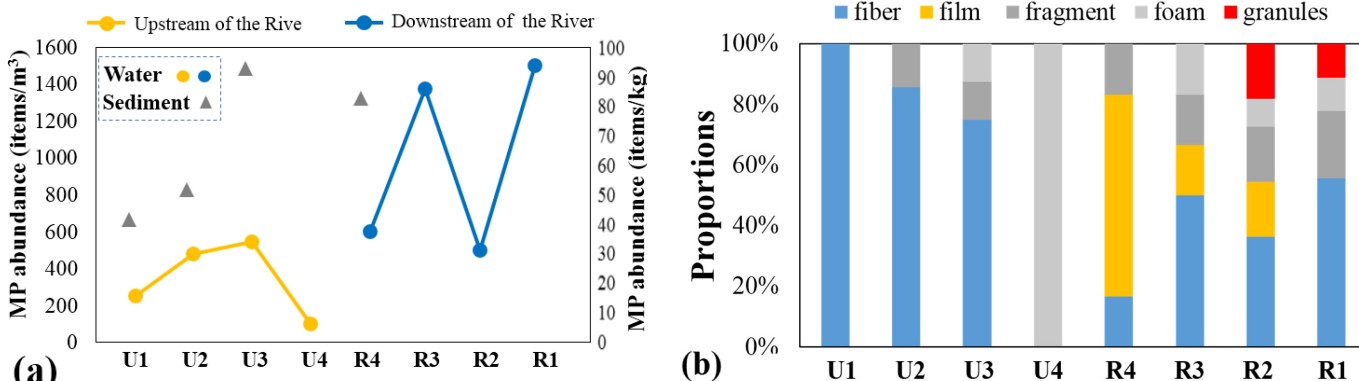

**Figure 5.** Abundance (**a**) and composition characteristics (**b**) of MPs in different sampling sites from the upper and lower reaches of the Brahmaputra River. Some data (U1, U2, U3, and U4 sites) from [31].

Furthermore, the different characteristics of MPs between studied upstream and downstream sections of the Brahmaputra River were investigated (Figure 5b) to study the MP sources in different environments in the study area. MP types in the downstream areas (R1, R2, R3, and R4 sites) were more diverse than in the upstream areas of the river (U1, U2, U3, and U4 sites) due to the variety sources of plastic waste input downstream. No granules were found in the more pristine and rural upstream areas, while a considerable number of granules was found in downstream samples. This hints at a strong connection between granule input and human activity in the downstream parts of the river, as granule occurrence has been linked to increased activities of daily human life [54]. In addition, as the largest tributary of Brahmaputra River, the average concentration of MPs in the Lhasa River reached 738 item/m$^3$, with a maximum value of 1286 item/m$^3$. As expected, the released MPs from the Lhasa River increased the mean MP abundance in the surface water of the Brahmaputra River, highlighting that tributaries are a major source of MPs in the studied downstream sections of the Brahmaputra River.

### 4. Conclusions

This study provided information on MP distribution in a high-altitude basin of the Tibetan Plateau, and the composition and source of MPs between the upper and lower reaches of the Brahmaputra River were compared. The MP abundances in the basin ranged from 188 item/m$^3$ to 1500 item/m$^3$ (average 735 item/m$^3$) in water and 30–90 items/kg (average 53 items/kg) in sediment. In addition, the relatively higher MP concentrations

in water samples were found near the Lhasa City (1285 item/m$^3$) and in the downstream sections of the Brahmaputra River. The concentration of MPs in the lower reaches of the Brahmaputra River was significantly higher than that in the headwaters of the upper reaches. On the one hand, a higher population density in the downstream areas contributed more plastic waste input. On the other hand, MPs from tributaries flowing into the river cannot be ignored, especially from the first major tributary, i.e., the Lhasa River. Future long-term and more systematic monitoring of MPs in rivers and lakes of the Tibetan Plateau is strongly required to delineate MP sources with more detail and develop mitigation measures where necessary.

**Supplementary Materials:** The following are available online at https://www.mdpi.com/article/10.3390/w13202805/s1, Figure S1: Photographs of typical sampling site, Figure S2: (a) Shape, (b) size, (c) color and (c) polymer distribution of MPs between different water body types, Table S1: Information for each sampling site of surface water and sediment, Table S2: Polymers of MPs identified in the water of study area, Text S1: Selection of separation fluid.

**Author Contributions:** S.F.: Conceptualization, Methodology, Writing–original draft. H.L.: Review, Supervision, Writing-review & editing. T.Y.: Conceptualization, Methodology. All authors have read and agreed to the published version of the manuscript

**Funding:** This study was funded by the Second Tibetan Plateau Scientific Expedition and Research Program (STEP) (Grant No. 2019QZKK1003), Strategic Priority Research Program of Chinese Academy of Sciences (XDA20040301), National Key Research and Development Program of China (Grant No.2019YFC0507801), and the CAS Interdisciplinary Innovation Team (Grant No. JCTD-2019-04).

**Institutional Review Board Statement:** Not applicable

**Informed Consent Statement:** Not applicable.

**Data Availability Statement:** The data presented in this study are available on request from the corresponding authors.

**Acknowledgments:** This study was supported by the Second Tibetan Plateau Scientific Expedition and Research Program (STEP) (Grant No. 2019QZKK1003), Strategic Priority Research Program of Chinese Academy of Sciences (XDA20040301), National Key Research and Development Program of China (Grant No.2019YFC0507801), and the CAS Interdisciplinary Innovation Team (Grant No. JCTD-2019-04).

**Conflicts of Interest:** The authors declare no conflict of interest.

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
