# Peer review of "Microplastics Footprints in a High-Altitude Basin of the Tibetan Plateau, China"

_water, doi:10.3390/w13202805_

Round 1

Reviewer 1 Report

Please find my comments in the attached pdf.

Author Response

Please see the attachment for specific replies

Reviewer 2 Report

line 81: passes, not is passess, receives, not receive
line 78-89 should be corrected, according to the English grammar
in Figure 1 the meaning of the red star is not explained, and it seems there are not any urban boundaries
Please explain why the stainless-steel screen was used for sampling instead of neuston nets, how the sampling was achieved in detail?

Reviewer 3 Report

Diffuse occurrence and steady increase of microplastics in freshwater ecosystems represent an important environmental problem. The present paper provides some interesting data, collected in the Tibetan Plateau, on the occurrence of microplastics in remote and impacted freshwater habitats. However, there are some major and minor issues (listed below) that should be fixed before publication.

Major:

  • The language needs to be substantially improved by a native or a professional with an understanding of the topic of the article. As an example, I provide the revised text of a particularly important section of the paper (Abstract):

“Microplastics (MPs) are ubiquitous in the environment and have been drawing increasing attention. However, MPs’ occurrence and behavior in remote areas are not well understood. In this study, we quantified and characterized MPs from surface waters and sediments in a remote area, namely the Tibetan Plateau, China. The samples were collected from the Lhasa River and the lower reaches of the Yarlung Zangbo River to better understand MPs’ sources to rivers of the Tibetan Plateau. MPs’ concentrations in water and sediment were 735 items/m3 and 51 items/kg, respectively, and the dominating MPs observed were fibers with size ranging from 100 to 500 μm. MP abundance increased nearly two times of magnitude from upstream to downstream in the Yarlung Zangbo River, associated with the inputs from downstream human activities and the inflows of tributaries (especially the Lhasa River). This study provides important bases for analyzing MPs migration processes in the plateau region.”

  • Methods: Numbers and types of samples need to be clearly listed. It should be straightforward for the reader to understand how many non-replicate water and sediment samples were taken.
  • Methods: Did you really treat the samples with hydrogen peroxide for 72 hours (3 days!). This seems an exceedingly long duration of sample preparation…
  • Methods: An anova with two groups is considered a t-test. Did you carry out an anova or t-tests?
  • Literature suggestions: I think that the following two articles might provide some further useful data for comparisons: * First insights into plastic and microplastic occurrence in biotic and abiotic compartments, and snow from a high-mountain lake (Carnic Alps). Chemosphere. DOI: 10.1016/j.chemosphere.2020.129121 * The fate of microplastics in an Italian Wastewater Treatment Plant. Science of the Total Environment. DOI: 10.1016/j.scitotenv.2018.10.269

Minor:

  • For the MPs items, please apply roundings to avoid decimal digits, e.g. change “743.9 ± 382.8 items/m3” to “744 ± 383 items/m3”.
  • 4: Correct to “3. Results and Discussion”.
  • Table 1: Correct to “Africa”.
  • 4: “p < 0.05”. “p” should be in Italics.
  • 7, line 235: Correct to “river”.
  • 8, lines 273-274: Fix line break.

Round 2

Reviewer 1 Report

Dear authors,

I have revised your improved manuscript and think it is much better than the first version. However, I found a number of unclear sentences and wording throughout the text, which makes it difficult to read at times. 

I have attached a my suggestions for improving these sentences and hope I understood correctly what you wanted to say. I think you are nearly there and my proposed corrections can be fixed easily. 

Author Response

We are thankful for the reviewer’s careful review and helpful advices, and have modified and/or corrected the context as the reviewer’s suggestions as shown in the new manuscript.  In addition, we have answered some questions separately in the attachment.

Reviewer 3 Report

In my opinion, the manuscript has been sufficiently improved to warrant publication in Water

Author Response

 We are thankful for the reviewer’s recognition.